# Analyzing Spatiotemporal Development of Organic Farming in Poland

**Elżbieta Antczak** 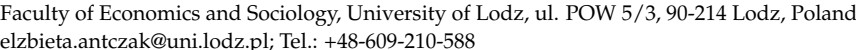

Faculty of Economics and Sociology, University of Lodz, ul. POW 5/3, 90-214 Lodz, Poland;
elzbieta.antczak@uni.lodz.pl; Tel.: +48-609-210-588

**Abstract:** Organic farming is one of the most widely known sustainable models of agricultural production. Success in eco-agriculture also depends greatly on agri-environmental, territorial, economic, social, institutional and spatial conditions. Polish eco-farming is very regionally dispersed and diversified. Regarding the important contribution of organic farming, a better understanding of how this phenomenon develops and which factors affect its spatial distribution can be influential for policymakers in planning strategies that pursue sustainable development objectives in rural areas. This paper assesses the development and analyses the spatial distribution of organic farming in Polish LAU-2. The country's eco-agriculture was mapped and defined using a synthetic measure, described by 27 sub-measurements of ecological crop cultivation, animal maintenance and eco-production. The local spatial patterns (direction, scale, and range) of organic farming were detected by spatial autocorrelation measurements. The analysis was conducted for the period 2014–2020. Possible external and internal determinants of this spatial dispersion were also defined. The results indicate that the distribution and spread of organic farming in Poland are related to public support, institutional regulations, social considerations, environmental concerns, the local job market and spatial dependencies.

**Keywords:** Polish organic farming; spatial disparities and dependencies; synthetic measure; local socio-economic factors





## 1. Introduction

Organic (ecological) farming is an agriculture system characterized by sustainable crop and animal production which ought to combine environmentally friendly practices, support high biodiversity, take advantage of natural processes and ensure animal well-being [1]. Organic farming, as a relatively novel and competitive agricultural production method, encounters numerous barriers and various limitations on its development pathway [2]. Major determinants of ecological production include: financial factors connected with the availability of support and subsidies (including compensations for agricultural losses) [3]; environmental conditions associated with biodiversity, landform features, and soil fertility, among others [4]; market-related aspects that arise from eco-product prices compared to conventional production and availability of organic products [5]; legal, systemic and institutional factors that connect farms and the market [6]; social factors associated with the labor force, stemming from lifestyle changes and social awareness [7]; regional factors resulting from the agrarian structure, character of the region (industrial or agricultural) and local historical determinants [8]; or spatiality connected with spatial dependencies, or the concentration or dispersion of eco-agriculture [9].

Eco-agricultural production is of increasing importance in Poland, where interest in organic farming significantly grew in the early 1990s [10]. A particularly advantageous time in Poland was its accession to the European Union (after 2004), which created favorable conditions for ecological production development thanks to subsidizing opportunities and the opening of external markets [11]. However, for several years, Polish organic farming has experienced stagnation: the number of producers fell from 25,000 in 2014 to 20,000

in 2020, while the crop surface area shrank from 670,000 ha in 2014 to about 510,000 ha in 2020. Nevertheless, since 2014, the number of food-processing plants has doubled, and ecological crop output has grown in almost all cultivation groups [12]. Although Polish consumers' annual expenditure on organic products amounts to less than one-tenth of the European Union (EU) average (around 6 euros per capita), and the share of ecological crops in Poland accounts for merely 3.5% (the EU-27 average being 8.5%), Poland is considered a country with high organic farming development potential [13]. In 2020, the surface area of ecological crops was 0.5m ha, while the organic food segment value increased to 245 m euros (20% more than in 2019) [14]. Moreover, Poland has considerable resources of arable land at its disposal. Among the EU countries, only France and Spain have more farmland, while comparable arable land resources are found in the United Kingdom, Germany, Romania, and Italy [15]. Although ecological agriculture constitutes an important element of Poland's economy, it is characterized by significant fragmentation of organic food production and supply. The surface area of most organic farms in Poland does not exceed 50 ha (the average farm has 22 ha, while in France, for example, it has 60.9 ha and in Germany—60.5 ha) [16]. The dispersion and considerable fragmentation of ecological agriculture reduce the chances of maintaining its competitiveness in the long run, weaken market accessibility outside large conurbations, slow down capital accumulation and generate low labor costs compared to Western European countries.

Considering the substantial contribution of organic farming in Poland, it can be crucial to policymakers in planning strategies that pursue sustainable development objectives in rural areas to better understand how the phenomenon develops and which factors affect its spatial distribution. Most Polish studies into organic farming focus on analyzing eco-agriculture conditions [17–21], selected efficiency aspects [22,23], farming type comparisons and development outlooks [24,25]. The results of the studies highlight economic and natural conditions that are favorable to the development of organic production in Poland. Some studies explain the barriers to the development of eco-farming [6,26,27]; however, as regards the assessment of the development capacity of organic farming, the prevailing opinion is that development opportunities are greater than factors that threaten the development trend. Many studies also concern the issue of supporting ecological agriculture, pointing to the fact that the system of subsidies was the major factor behind the fast rise in the number of organic farms, although it did not stimulate simultaneous increases in production and processing [28,29]. With the growing body of foreign literature (e.g., [4,30,31]), the spatial context of organic farming (spatial autocorrelation, local dispersion and disparities) has been relatively neglected in Polish research.

This study is the first attempt at analyzing the spatiality (spatial and temporal patterns, spatial dependencies) of organic farming development in Polish communes. The paper defines eco-agriculture using a dynamic, synthetic measure. Local spatial patterns (direction, scale and range) of organic farming are detected through spatial autocorrelation measurements (global and local Moran's I indices). The analysis is conducted for the years 2014–2020. To better capture and recognize regional differences among communes, the results of the analysis are mapped, and possible determinants of spatial distribution are also defined. Thus, this paper contributes to the literature in several dimensions. Although the topic has been broadly researched at the country level, studies that address sub-regional units are still scarce. Hence, this research comprehensively investigates the spatial dispersion of eco-agriculture development at the local level in Poland. The analysis covers all Polish communes (N = 2477). The data concern communes, as they are the smallest geographical units responsible for shaping organic farming development through integrating tasks related to the protection of biodiversity, historical and landscape aspects, and community living conditions. Moreover, official Polish statistics already give an overview of organic farming at the national level, but organic crop farmers are smallholders located in urban or rural areas. To assess organic farming development and analyze its spatial distribution, unique data were obtained at the lowest possible spatial aggregation level (from the Main Inspectorate of Agricultural and Food Quality Inspection, MIAFQI). Draw-

ing on raw data describing the harvest level, animal headage, crop amount and animal output, 27 indices were constructed that characterize organic farming in Poland. Based on those indices, a synthetic, dynamic measure of ecological agriculture development was devised. Finally, by analyzing the spatial autocorrelation, a new paradigm of Polish organic farming was defined. Identifying the areas where organic farming is present today and understanding its spatial dispersion is crucial to improving access to organic certification and supporting the expansion of certified cropland in the future. Moreover, to be fully aware of the potential benefits and consequences of organic farming, we need to understand its intersections with local environments and cultures, as well as control spatial dependencies (because of strategic interactions, indirect effects of exogenous factors, or spatial correlations in the environment in which decision-makers operate). To investigate the extent to which spatial interactions take place, different spatial weights matrices were used. Such a novel and multifaceted approach has not previously been employed, and its results should be relevant to formulating agriculture policy recommendations, especially for small-sized, regionally diversified economies.

## 2. Materials and Methods

### 2.1. Materials

Poland is divided into three regional classification levels (NUTS) and two levels of Local Administrative Units (LAU-1—counties, and LAU-2—communes). As of 1 January 2021, there were 2477 communes (NUTS-5 or LAU-2) [32]. To evaluate the condition and spatial variation of organic farming in Polish communes, 27 diagnostic variables (indicators) were used (designated based on data from MIAFQI [33] and the Central Statistical Office (CSO) [34]). The data represent four areas: entities, plants, animals, and products. The analysis covers the 2014–2020 period. The time range of the data was narrowed down to the above years due to the uniform way databases are building and regularly updated at the commune level, containing similar descriptions of organic farm features, unified production categories and the potential impact of support obtained from EU funds in the framework of the Rural Development Program for the years 2014–2020 at the production level [35]. The diagnostic variables were assessed with high space variation [36] and relatively low correlation [37]. Ultimately, 25 characteristics met the criteria and were taken into consideration in the dynamic synthetic measure (4)—variables not tinted grey, Table 1.

**Table 1.** Diagnostic variables of organic farming and descriptive statistics (averaged over the years 2014–2020).

| Diagnostic Variables | Av. | CV [%] | AC: [% or pp] | GM | Include in DSM |
|---|---|---|---|---|---|
| **Entities** | | | | | |
| Organic producers [per 1000 entities entered in the national official register] | 17.4 | 303 | −39 | 0.67 *** | No |
| Share of the organic agricultural area [% of the total agricultural area] | 3.1 | 260 | −33 | 0.22 *** | No |
| **Plants** | | | | | |
| Yields of cereals (maize, oats, barley, rye, triticale, wheat) grown for grain (including seed) [kg per capita] | 11.1 | 296 | 120 | 0.36 *** | Yes |
| Dry bean harvest [kg per inhabitant] | 1.5 | 368 | 281 | 0.35 *** | Yes |
| Harvest of root crops (potatoes, sugar beet and other) [kg per capita] | 1.3 | 495 | 14 | 0.20 *** | Yes |

**Table 1.** *Cont.*

| Diagnostic Variables | Av. | CV [%] | AC: [% or pp] | GM | Include in DSM |
|---|---|---|---|---|---|
| **Plants** | | | | | |
| Harvest of industrial plants (hops, rape, colza, sunflower, soybean, flax, medicinal and spice plants) [kg per capita] | 0.6 | 1322 | 984 | 0.01 ** | Yes |
| Harvest of vegetables (brassica, leaf, stem, onion, root, peas, beans, mushrooms) [kg per capita] | 2.4 | 588 | 267 | 0.16 *** | Yes |
| Harvest of strawberries and wild strawberries [kg per capita] | 0.4 | 860 | 53 | 0.15 *** | Yes |
| Fodder crops (maize, fodder beet, dicotyledonous, grass) [tons per hectare of organic area] | 1.5 | 138 | −14 | 0.42 *** | Yes |
| Harvest of crops from seed plantations [tons per hectare of organic area] | 4.9 | 2423 | −99 | −0.02 | Yes |
| Harvest from fruit trees and shrubs (fruit and berry crops) [kg per capita] | 5 | 643 | 260 | 0.18 *** | Yes |
| Harvest from vineyards [kg per inhabitant] | 0.04 | 2552 | 1005 | 0.001 | Yes |
| Harvest of flowers and ornamental plants [kg per capita] | 0.001 | 4876 | −100 | −0.001 | Yes |
| **Animals** | | | | | |
| The number of cattle kept for meat and milk [per 1000 population] | 1.8 | 583 | −5 | 0.11 *** | Yes |
| The number of pigs (fatteners, sows) [per 1000 population] | 0.3 | 816 | −65 | 0.05 * | Yes |
| Sheep (ewes and others) [per 1000 population] | 1 | 654 | −43 | 0.09 ** | Yes |
| Headcount of goats (mothers and others) [per 1000 population] | 0.2 | 678 | −21 | 0.04 *** | Yes |
| Number of rabbits (female and other) [per 1000 population] | 0.4 | 2822 | 96 | −0.001 | Yes |
| Poultry (broilers, chickens, ducks, turkeys, geese, ostriches) [per 1000 population] | 15.9 | 654 | 102 | 0.02 ** | Yes |
| The number of horses (equines) [per 1000 population] | 0.04 | 534 | 164 | 0.07 *** | Yes |
| The number of deer (noble and sika) and fallow deer [per 1000 population] | 0.2 | 1740 | −76 | 0.01** | Yes |
| The number of snails [kg per capita] | 0.01 | 6300 | 100 | −0.001 | Yes |
| **Products** | | | | | |
| Production of milk and cream [litres per capita] | 1.2 | 1203 | 55 | 0.06 ** | Yes |
| Production of butter, cheese [kg per capita] | 220.7 | 1313 | 55 | 0.05 ** | Yes |
| Egg production (including eggs for consumption) [number per capita] | 2.3 | 943 | 327 | 0.03 ** | Yes |
| Meat production [kg per capita] | 0.001 | 2659 | −29 | −0.01 | Yes |
| Honey production [kg per capita] | 0.04 | 4797 | 1500 | −0.01 | Yes |

Note: among the 27 organic farming designated variables, all were stimulants, i.e., their high values are favorable for the studied phenomenon [38]. The eco-farming of seaweeds and fishery was not observed in Poland; pp—percentage point; CV—coefficient of variation; Av.—average; AC—change of average value in 2020 in relation to 2014; GM—global Moran's I; DSM—the dynamic synthetic measure; significance levels: * $\alpha = 0.10$, ** $\alpha = 0.05$; *** $\alpha = 0.01$; the fourth order queen criteria and row standardized spatial matrix was used [39].

### 2.2. Methods

2.2.1. Dynamic Synthetic Measure

To assess the state and spatial variability of organic farming, the dynamic synthetic measure ($DSM_{it}$) was built for all communes in the period 2014–2020 using the zero unitarization method [40]. The method assumptions and its subsequent stages included:

1.  Presenting the diagnostic variable of organic farming (Table 1) $X_j$ ($j = 1, 2, \ldots, m$) for each commune $Oi$ ($i = 1, 2, \ldots, n$) in each studied period in the form of a two-dimensional matrix (1):

$$X = \begin{bmatrix} x_{11} & x_{12} & \dots & x_{1m} \\ x_{21} & x_{22} & \dots & x_{2m} \\ \vdots & \vdots & \vdots & \vdots \\ x_{n1} & x_{n2} & \dots & x_{nm} \end{bmatrix}. \tag{1}$$

2.  Conducting preliminary correlation and variability analysis to exclude variables due to their strong association with each other and low degree of variability. Pearson's linear correlation coefficient was adopted to measure the strength and direction of the correlation between the observed variables (the Student's *t*-test for significance of correlation was also applied) [36]. Variability was expressed by the coefficient of variation (CV), which is generally claimed to be more than 10% [37].

3.  Normalizing the variables to maintain comparability of statistical data. The stimulants are normalized with the Formula (2):

$$z_{ijt} = \frac{x_{ijt} - min\{x_{ijt}\}}{max\{x_{ijt}\} - min\{x_{ijt}\}} \ (i = 1, 2, \dots, n); \ (j = 1, 2, \dots, m); \ (t = 1, 2, \dots, l) \ z_{ijt} \in [0,1] \tag{2}$$

and the destimulants with the Formula (3):

$$z_{ijt} = \frac{man\{x_{ijt}\} - x_{ijt}}{max\{x_{ijt}\} - min\{x_{ijt}\}} \ (i = 1, 2, \dots, n); \ (j = 1, 2, \dots, m); \ (t = 1, 2, \dots, l) \ z_{ijt} \in [0,1], \tag{3}$$

where: $z_{ijt}$—the normalized value of the *j*th variable for the *i*th object and *t*th period; $x_{ijt}$—the value of the *j*th variable for the *i*th object and the *t*th period; $maxx_{ijt}$—the maximum value of the *j*th variable for all *i*th objects and all *t*th periods; $minx_{ijt}$—minimum value of the *j*th variable for all *i*th objects and all *t*th periods [41]. In a basic (static) version of the scaling, $maxx_{ijt}$ and $minx_{ijt}$ are the maximum and minimum values of variable *x* for a given time unit. In a dynamic approach, however, the maximum and minimum values for all objects and all-time units are selected, and the values of the normalized variables still go beyond the interval [0, 1]. [42].

4.  Calculating the dynamic synthetic measure ($DSM_{it}$) as an arithmetical mean of normalized (2) and (3) variable values (4):

$$DSM_{it} = \frac{1}{m} \sum_{j=1}^{m} z_{ijt}, \ (i = 1, \dots, n); (j = 1, \dots, m); (t = 1, \dots, l) \tag{4}$$

The dynamic synthetic measure obtained through Formula (4) assumes values in the interval [0, 1]. This method makes it possible to rank the communes with the best (close to 1) and the worst (close to 0) levels of organic farming, whereas it would be difficult to identify organic farming leaders and followers based on individual indicators.

5.  Map visualization, which plays a key role in interpreting the results of variability and understanding the state and development of organic farming from a spatiotemporal perspective.

ArcMap software was applied at this stage of the analysis.

### 2.2.2. Spatial Autocorrelation

Spatial autocorrelation means that nearby units are more likely to be related than more distant ones [43]. To assess the spatial dependencies and spatial patterns in organic farming among Polish communes, spatial autocorrelation Moran's I indices were applied [44]. The global Moran's index (5) was adopted to detect the global spatial interactions, while Moran local autocorrelation index (LISA) (6) measured the degree of spatial autocorrelation at each specific site:

$$I = \frac{n \sum_{i=1}^{n} \sum_{j=1}^{n} w_{ij}(x_i - \bar{x})(x_j - \bar{x})}{s_0 \sum_{i=1}^{n}(x_i - \bar{x})^2}, \tag{5}$$

where: $n$—the number of spatial units (communes); $x_i$ and $x_j$ —the values of attribute $X$ considered in areas $i$ and $j$; $\bar{x}$—the average value of attribute $X$ in the studied commune; $w_{ij}$—the element of the normalized neighborhood matrix, corresponding to the spatial weights 0 and 1, with "0" for areas $i$ and $j$ that do not border each other and "1" for communes $i$ and $j$ that do border each other; $s_0$—the sum of the elements $w_{ij}$ of the symmetrical spatial weights matrix $W$, which is $\sum\limits_{i=1}^{n}\sum\limits_{j=1}^{n} w_{ij}$.

The value of Moran's statistics varies from $-1$ to 1. Positive spatial autocorrelation means that locations close together have similar values, while negative spatial autocorrelation means that locations close together have more dissimilar values than those that are further away [45].

The LISA, based on the global Moran's index, identifies local patterns of spatial fragmentation and association and extreme spatial values [46]:

$$I_i = \frac{x_i - \mu}{\sigma_n^2} \sum_{j=1}^{n} w_{ij}(x_j - \mu), \ i = 1, \ldots, n, \tag{6}$$

where: $\sigma_n^2$—the variance of variable $X$ being studied in $n$ communes $\sigma_n^2 = \frac{\sum_{i=1}^{n}(x_i - \mu)^2}{n}$; $x_i$—the observation of variable of $X$ in commune $i$ for $i = 1, \ldots, n$; $\mu$—the average of $n$ communes.

To verify hypotheses concerning spatial autocorrelation, randomization tests are performed [44]. The spatial autocorrelation analysis was carried out in GeoDa.

## 3. Results

### 3.1. Preliminary Data Analysis

Table 1 shows a 39% fall in the number of organic farms per 1000 economic entities entered in the REGON National Economic Register between 2014 and 2020. The share of organic farming surface area in arable land accounted for 3.4% and saw a 33 pp drop over the period. Communes were also characterized by great variation in both the number and the surface area of organic farms (CV > 10%). The crop production volumes also varied among communes (CV ranging from 138% to 4876%). In the period 2014–2020, the predominant crops were cereals grown for grain (including sowable material, i.e., maize, oats, barley, rye, triticale and wheat). Mean cereal crops were 11 kg per capita, corresponding to a rise of 120%. Orchard and berry cultivation also recorded considerable crops, at approximately 5 kg per capita with a simultaneous increase of roughly 260%. Meanwhile, vineyard crop volumes saw the largest rise—of more than 1000%—and the greatest degree of spatial variation in communes. Drops in organic crops were noted for flowers and ornamental plants, fodder plants and sowable material plantations.

In turn, poultry breeding saw the largest scale of animal eco-production. Between 2014 and 2020, the average quantity of poultry, including broiler chickens, laying hens, ducks, turkeys, geese and ostriches, was about 16 kg per capita, with a simultaneous headage increase of 102%. However, the highest growth was noted in the headage of horses, 164%. In turn, a considerable drop in the ecological breeding of animals was observed for the headage of pigs (porkers, sows), i.e., of 65%, as well as the headage of deer (red deer and sika deer) and European fallow deer, i.e., of 76%, for which significant commune variation was also noted (CV = 1740%). A more than 10-fold increase in organic honey production was observed, although its average quantity per capita was 0.04 kg. Meanwhile, cheese and butter had the highest output at more than 220 kg per capita, Table 1.

### 3.2. Synthetic Measure of Organic Farming

A rise in organic farming output occurred in communes between 2014 and 2020. The synthetic measure value grew from 0.013 of the unit in 2014 to 0.021 in 2020, with an annual average of 7% growth (see Figure 1). Especially considerable agricultural development was noted in the final two years (a maximum measure value of 0.71 of the unit in 2019 and

0.84 in 2020, compared to a maximum of 0.45 in 2014). In light of the results, it may also be stated that Poland was characterized by considerable commune variation of organic farming output (see Figure 1). Additionally, there was a drop in the number of communes in which organic farming was run in the studied period (from 2022 units in 2014 to 1885 in 2020). High levels and growing intensity of organic farming may be observed in communes situated in the north-eastern, north-western, eastern and south-eastern parts of Poland. The numbers of units dealing with organic farming rose in eastern and north-eastern Poland, whereas there was a marked drop in the numbers of such communes in the west and south of Poland. The analysis indicates that between 2014 and 2020, the highest positions in the ranking of organic farming development were taken by communes in the following order: Godkowo—located in the north-eastern part of Poland (almost all of which consists of the Protected Landscape Areas of the Pasłęka River along with the "Beavers on the Pasłęka River" and the Wąska River reserves) [24], Krempna—which is an important center of organic farming development in south-eastern Poland, Biały Bór—in north-west Poland, and Tarnogród in the east. Over the studied period, the Godkowo commune showed the most abundant cereal crops, in particular, common wheat, rye and oat, pulses, potatoes, and root and bulb vegetables (especially carrots). Moreover, the commune specializes in the breeding of sheep, sows and geese, as well as cattle kept for meat and milk cows. In turn, the Krempna commune—despite scarce human resources in the local labor market and lack of local authorities' support [47]—specializes in the growing of bulb and root plants (mainly potatoes) and fodder plants (grass on arable land), but its greatest potential is the breeding of pigs (mostly porkers) and sheep, as well as production of (cow's) milk and butter. In Biały Bór, more than 90% of the area is arable land, forests and woodlands, conditions that significantly exceed the mean national determinants of organic farming development [48]. While the commune boasts above-average crops of pulses for dry grain and is developing organic oat and rye production, it specializes mainly in the production of milk and sour milk. Large numbers of cattle kept for meat, milk cows, sheep and goats are also raised in the commune area. It is also worth drawing attention to the Biłgoraj commune, situated in the south-east. Although it is positioned somewhere in the middle of the ranking, it specializes in growing flowers, ornamental plants and herbs—the only commune of its type in Poland. It has very good soil conditions, favorable for even the most demanding crops [47]. The lowest positions in the ranking are held by communes located in central and south-western Poland, which are mainly urban centers (see Figure 1). The results are also interesting for communes where organic farming is registered, but no agricultural production took place between 2014 and 2020. As can be seen from the maps in Figure 1, most units of this type were recorded in 2014, 2015 and 2018.

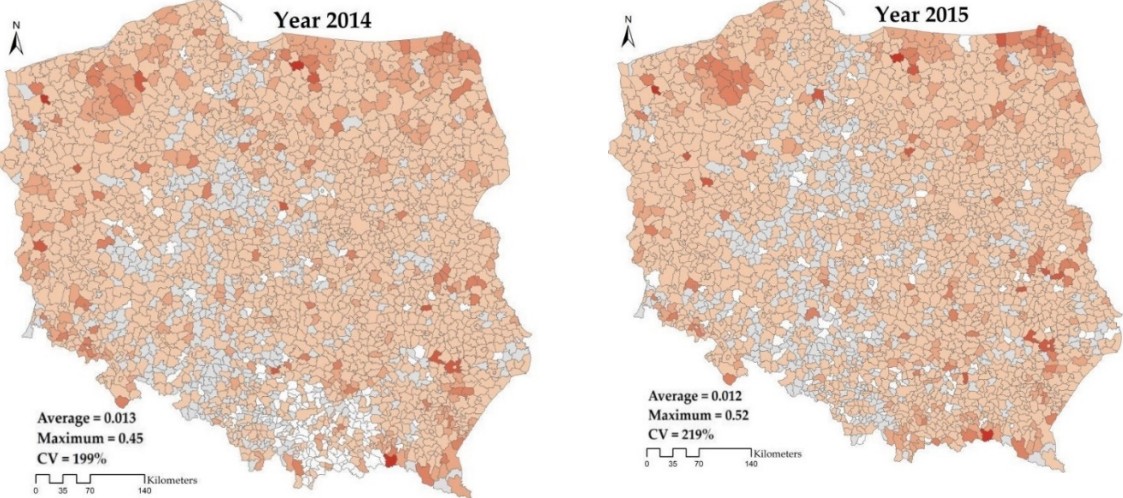

**Figure 1.** *Cont.*

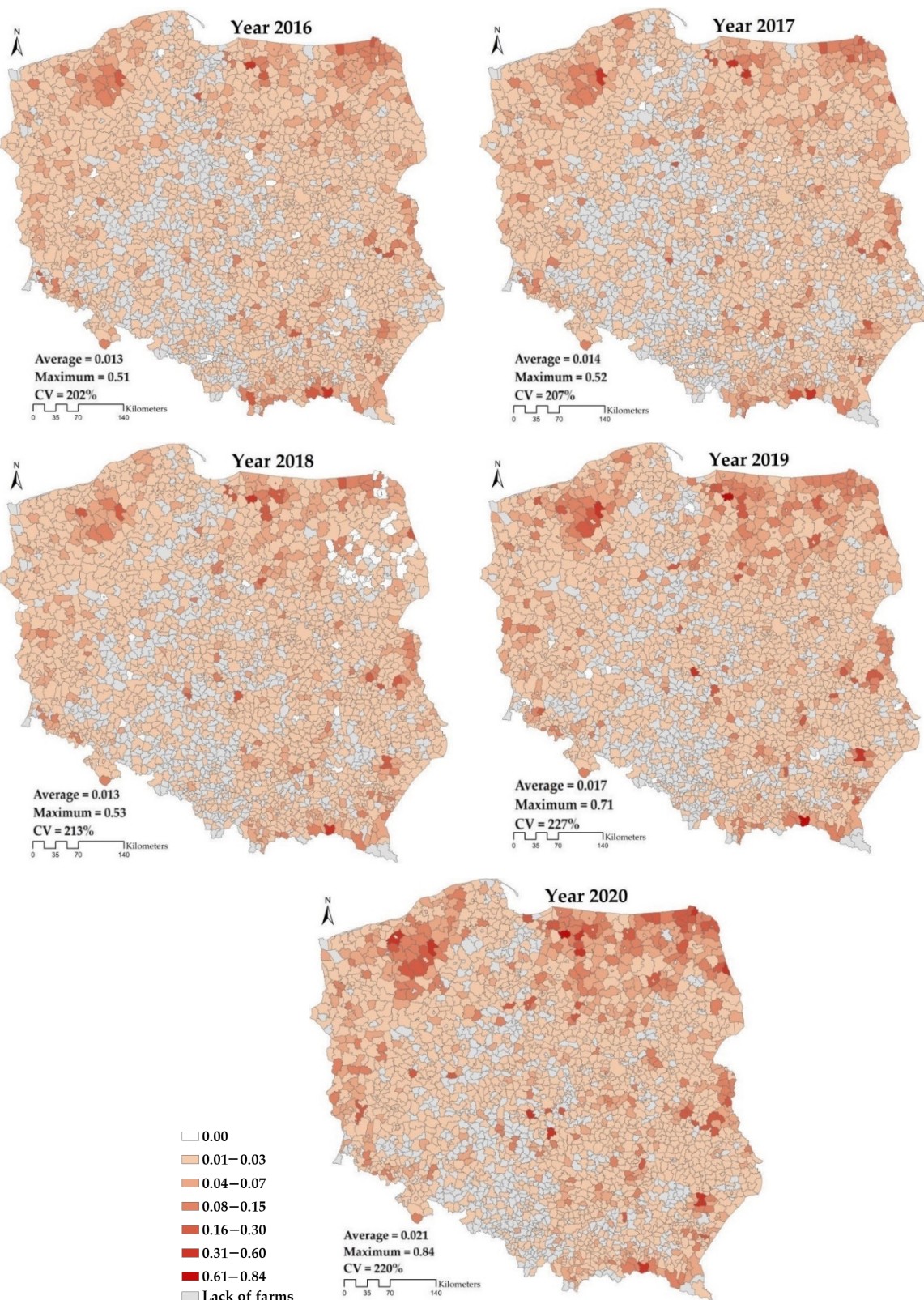

**Figure 1.** Synthetic measure of organic farming in Polish communes, 2014—2020. Note: for a robustness check of the results, a taxonomic development measure based on Hellwig's approach was applied [49]. If the synthetic measure of organic farming of Polish communes is applied to the data set, there is the division presented above that satisfies Hellwig's results and simultaneously corresponds to a particular object grouping in the illustrated datasets. The outcomes are available upon request.

### 3.3. Spatial Autocorrelation in Ecofarming

In organic agriculture, production in a focal unit can contribute to agricultural productivity in a proximate commune. These interdependencies may be the result of similar resources and human capital [50], spillovers from public agricultural research and development of public infrastructure [51], policy reform implications and similar conditions of the natural environment [52], targeted sectoral agriculture funding programs [53] or other indirect effects of exogenous factors (i.e., not under decision-makers' control) [54]. Attention to spatial interactions may lead to insights that would have been otherwise overlooked, while ignoring spatial dependencies between neighboring communes may result in biased estimations and false conclusions about the detected spatial patterns [55]. To identify and accurately describe spatial patterns of organic farming development, the global local Moran's statistics were applied (Equations (5) and (6)). To deal with potential inaccuracies (such as rounding errors) in the irregular polygons—like most areal units encountered in practice (including Polish communes [56]) using the queen criterion is recommended in practice. The neighbors in the queen criterion are units that have at least one point in common, including borders and corners [46]. The results of the analysis, shown in Table A1 (see in the Appendix A), confirm significant spatial processes that shape area-related organic farming development in communes. On the one hand, autocorrelation intensity decreased with distance until spatial dependencies petered out or transformed into spatial polarization. Moreover, growing spatial concentration intensity in time was observed, while larger-scale commune clustering concerned units of low agricultural development (low-low clusters) (see Figure 2). From one year to another, there was also a rise in the number of communes where spatial autocorrelation significantly affected organic farming development. It was also found that statistically significant spatial relationships reached as far as the 30th order of contiguity. The more in-depth analysis of the potential determinants of organic farming development's spatial patterns used the fourth order of contiguity matrix (also considering lower-order neighbors) (see Table A1). Thus, the area-related structure of natural conditions, including complexes of agricultural usefulness of soils, was reflected as accurately as possible [57].

The spatial interactions differentiate regions of Poland in terms of organic farming development. Particularly strong and prominent spatial trends towards commune clustering might be observed in the organic farming of fodder plants, cereals, beans, bulb, and root plants, as well as goat, sheep and horse breeding (Table 1). In turn, local spatial regimes (Figure 2) reflect various factors that shape the ecological agriculture potentials of communes. A prominent high-high commune cluster is situated in north-western Poland. In turn, the spatial structure of the commune cluster in north-eastern Poland with highly developed organic farming did not undergo noticeable changes. Nevertheless, the increasing intensity of the relationships was observed (with a high level of statistically significant clustering at $p = 0.01$), which consequently reinforced the already established ecological agriculture potential of the region's communes. Strong spatial dependencies were also observed in some communes situated in the east and south-east of the country. Meanwhile, the cluster of communes situated in the west of Poland was characterized by an observable variation of spatial processes that impact organic farming development (petering out spatial relationships were noted there between 2015 and 2019).

The period 2014–2020 also saw the formation of a prominent cluster of spatial dependencies among communes with low organic farming development levels (low-low cluster) running lengthwise across the country. Finally, numerous communes may also be indicated as significantly outlying the other communes around them, i.e., of untypical feature values (low-high and high-low), thus associated with locally negative spatial autocorrelations. The communes of central Poland are worthy of particular attention, showing a high level of organic farming development, but surrounded by units characterized by low variable values (high-low LISA values), e.g., the aforementioned Biłgoraj commune (see Section 3.2).

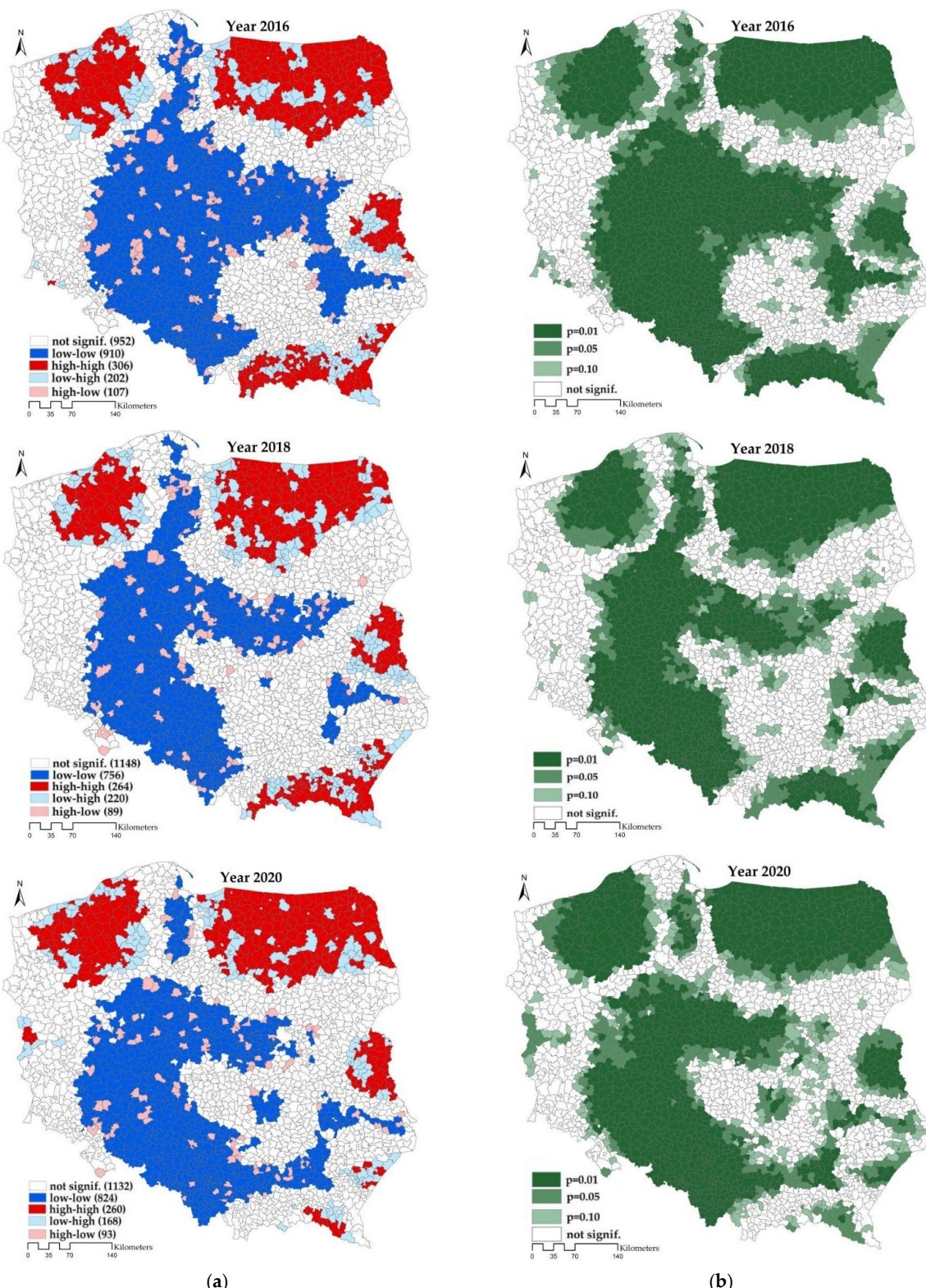

**Figure 2.** Local spatial clustering of organic farming in selected years, (**a**) cluster map and (**b**) cluster significance. Note: *p* are pseudo *p*-values calculated by determining the proportion of Local Moran's I values generated from permutations that display more clustering than the original data. If this proportion (the pseudo p-value) is small (less than 0.05), one can conclude that the data do display statistically significant clustering. Increasing the number of permutations increases precision by increasing the range of possible values for the pseudo-p. For example, with 99 permutations, the precision of the pseudo-*p* value is 0.01 [44].

## 4. Discussion

The results revealed a considerable commune variation and drop in the level of organic farming between 2014 and 2020 (Figure 1). That may indicate rising regional specialization and increasing local concentration of a particular organic farming line. A significant rise was observed in cereal crops cultivated for grain, bulb and root plants, and the headage of cattle bred for meat and milk, thanks to the increasing demand for Polish organic farming products, especially wheat and milk [58]. Paradoxically, the reason for the surge was the spread of coronavirus, and in particular, the lockdown, which determined the need to prepare and consume meals at home [59]. Simultaneously, there was growing interest in local, high-quality products [60]. The pandemic also relaxed legislation, introducing, for example, an opportunity for farmers to obtain additional financial assistance and making it possible to perform remote organic farming inspections using alternative methods and tools, such as Internet-based communication [61]. In general, since 2018, Polish law has stabilized (regulations concerning the mode and principles of aiding organic food producers changed as many as seven times between 2014 and 2018 [62]).

The research revealed enormous disparities in commune development related to the phenomenon but also made it possible to identify key Polish ecological agriculture development centers. The highest positions in the ranking of organic farming development were taken by clusters of communes located in north-western, north-eastern, south-eastern and eastern Poland. The development of organic farming in these regions displays many internal and external spatial factors. The meaningful organic farming crops in communes located in north-western Poland are determined by favorable local natural conditions, mainly rich soils and good water and climate conditions of the seaside and lake district areas [63]; lowland landscape [64]; favorable economic conditions in the form of the highest subsidies for ecological production in Poland in the framework of the Rural Development Program for 2007–2013 and 2014–2020 [65]; the proximity of the borderland and low level of industrialization [63]; relatively small distance to towns (not more than 50 km) [66]; and the quickly developing local fashion for so-called enotourism (wine tourism) [67]. Therefore, the region's communes specialize in growing cereals and pulses, industrial crops, including linseed, sunflower, medicinal plants and spices, and viticulture, as well as sheep and deer breeding. In turn, the north-eastern part of Poland has the most organic farms [68], which predominantly specialize in organic milk and butter production. The region's communes also show good crops of cereals, fodder plants and fruits–particularly apples and pears, as well as chokeberry. The production of honey and meat, and the raising of goats and poultry (mainly hens of the green-legged partridge breed) are also of importance to the region [69]. Strong spatial dependencies of that region's agriculture are determined by its great, above-average natural advantages, raw material base (peat and minerals) and highly natural character with the least polluted nature (numerous reserves, parks, lakes, the Green Lungs of Poland functional area) [70]. What also consolidates the ecological management of the region is farm tourism and weak industry—the popular stereotype of the region as the land of natural and tourist appeal does not encourage external investors to establish undertakings in the area [71]. Other factors that are favorable to organic farming development include the large supply of the labor force, especially in young people (18–34 years), which was determined by an above-average population increase between 1950 and 2015 [72], as well as the cooperation and numerous actions of local authorities to promote regional organic products [73]. Equally favorable natural features (in particular, soil quality) enable significant growth in organic food production [74] in eastern and south-eastern Poland. A high level of ecological agriculture concentrated in the region's communes is based on fruit plantations, mainly of berries (cowberry cultivation), growing strawberries and cereals, medicinal plants and spices, and vegetables. The communes also specialize in goat and rabbit breeding. Internal factors that determine the area's consistent organic farming development include the large labor force resources, fueled by migration from Ukraine and Belarus [75], considerable research and development potential [76] and growing ecological awareness. Those eastern regions of Poland, with their relatively low

costs of living, are becoming popular transregional academic destinations (many of the scientific institutions focus on environmental, agricultural and medical education and research) [74]. Another important issue connected with organic farming in the region is the noticeable support of the ecological agriculture production model provided by local self-government and organizations. They focus on promoting food production by organic farms and organizing training courses, conferences and events to promote and popularize the organic farming notion [77].

Meanwhile, the analysis revealed regional unstable shifts of organic farming over time in communes located in western Poland. This region's ecological agriculture focuses mainly on plant production–mostly of cereals (its share of cereals in the total area of crops is the highest in Poland), pulses, fodder plants, fruit trees (plums and cherries), chokeberry and currant. Since 2014, the area's natural assets have greatly improved—the vegetation period has become longer while emissions of air pollutants and soil contamination have decreased. Nevertheless, the conditions are still assessed as moderate for organic farming production. Agrarian structure improvement also contributes to area-related agriculture development— there are farms of relatively large surface area (from 10 to 50 ha) [78]. A rapid increase in production that applies organic methods was also encouraged by introducing considerable state financial support in the form of subsidies for farm inspection costs and farm surface areas [79].

The lowest level of phenomena is observed in suburban areas, where spatial planning is not sufficiently developed, which poses a constant challenge for the effective management of the rural landscape in the vicinity of cities [80]. Moreover, the low-low organic farming belt is formed by units—mainly urban centers—characterized by high industrial contamination, high population and infrastructure density [81], high environmental pollution and quick (sub)urbanization processes associated with taking over arable land for non-agricultural purposes [82]. Lower-class soils predominate in those communes [83], and the areas of the units are also at extreme risk of droughts [84]. Despite their below-national-average organic farming conditions, the regions play an important role in Poland's plant production, including potatoes, rye, fruits, vegetables grown in the open, honey, mixtures of cereals, sunflower, oat and maize, as well as pig breeding [85].

In turn, unprofitability and ceased organic production, especially for areas where organic farming is registered but no agricultural production took place, might have resulted from the lack of financial support or EU subsidies, or tightening requirements imposed on organic farms between 2013 and 2016 [86].

## 5. Conclusions

The aim of the research was to assess the spatial variation of organic farming development in Polish communes between 2014 and 2020. The synthetic measure made it possible to rank the units based on their level of development. In turn, the spatial autocorrelation analysis revealed spatial patterns—the area-related character of organic farming in the years studied. The change concerned not only the number of communes where organic farms operate, but also the spatial distribution (concentration) of organic farming. The analysis results indicated increasing interregional ecological agriculture variation with a simultaneous trend towards considerable local clustering of units in space and their potential specialization in organic farming. The changes taking place in the spatial structure was especially noticeable in communes of the north, south and west of the country. There was a substantial strengthening of the commune cluster position and spatial interactions in north-western Poland in 2014. The number of communes with a similar—high-level grew in the vicinity of units characterized by high eco-agriculture development levels (high-high LISA cluster). What took place was the so-called spillover of spatial dependencies, contributing to agricultural development. However, in central and south-central Poland, a cluster of communes with low organic-method production formed, while the number of outlying units diminished—especially those characterized by low eco-agriculture development and surrounded by communes with high development

levels (low-high outliers). The strength of spatial also dependencies increased (particularly relationships at $p = 0.01$).

The detection of spatial regimes (of different strengths and ranges) made it possible to identify factors that polarize regions and cluster communes in terms of development. Factors that determine the high level of organic farming development were: natural conditions—especially rich soils, naturally valuable areas of low pollution and highly natural character; the introduction and use of subsidies from the national and EU budgets; consistent and stable legal regulations; and the proximity of potential markets for goods in the form of large urban centers. In turn, internal factors that characterize particular groups of units were: non-agricultural farm tourism carried out by farms in the commune area; the cooperation of communes with local authorities; support by organizations and heads of communes in the form of promotional campaigns, fairs and conferences; the labor force; access to research and development centers; and growing ecological awareness of local communities. It is also interesting that, notwithstanding the world crisis caused by the consequences of the COVID-19 pandemic, it was in 2019 and 2020 that the fastest development of agriculture was observed in Polish communes.

The findings of this study also suggest significant methodological and practical implications for locally concentrated organic farming management and control. The results may be relevant to authorities seeking to implement better policies aimed at boosting eco-awareness and promoting national organic food products. Moreover, as this research focuses on the LAU-2 level and is publicly available, local creators could, therefore, easily access and use the outcomes for planning promotional strategies, better predicting distribution channels and more effectively supporting organic farming production.

The analysis points to the significant and long-lasting place of organic farming in Polish agriculture. However, the results also confirmed that the development of the organic farming system is unstable, spatially varied and multidirectional. Identifying the factors that determine the demand for organic foods and looking for outcomes of indicated spatial interactions will be the subject of further empirical research.

**Funding:** This research received no external funding.

**Informed Consent Statement:** Not applicable.

**Data Availability Statement:** The data presented in this study are openly available from the Main Inspectorate of Agricultural and Food Quality Inspection: https://www.gov.pl/web/ijhars/rolnictwo-ekologiczne (accessed on 16 September 2021) and from Polish Local Data Bank: https://bdl.stat.gov.pl/BDL/start (accessed on 16 September 2021).

**Conflicts of Interest:** I declare no conflict of interest. The funders had no role in the design of the study; in the collection, analyses, or interpretation of data; in the writing of the manuscript, or in the decision to publish the results.

## Appendix A

**Table A1.** Results of organic farming spatial autocorrelation using different spatial weights matrices.

| Order of Contiguity (Including Lower Order/s) | LISA Clusters/Outliers | DSM2020 | DSM2019 | DSM2018 | DSM2017 | DSM2016 | DSM2015 | DSM2014 |
|---|---|---|---|---|---|---|---|---|
| | Moran's I | 0.28 *** | 0.23 *** | 0.22 *** | 0.25 *** | 0.27 *** | 0.20 *** | 0.18 *** |
| | HH | 155 | 147 | 134 | 148 | 155 | 119 | 136 |
| 1st | LL | 348 | 361 | 296 | 352 | 383 | 311 | 306 |
| | LH | 55 | 59 | 61 | 49 | 49 | 73 | 67 |
| | HL | 22 | 38 | 39 | 30 | 36 | 22 | 29 |
| | Not sig. | 1987 | 1872 | 1947 | 1898 | 1854 | 1952 | 1939 |

**Table A1.** *Cont.*

| Order of Contiguity (Including Lower Order/s) | LISA Clusters/Outliers | DSM2020 | DSM2019 | DSM2018 | DSM2017 | DSM2016 | DSM2015 | DSM2014 |
|---|---|---|---|---|---|---|---|---|
| 2nd | Moran's I | 0.21 *** (0.23 ***) | 0.17 *** (0.19 ***) | 0.17 *** (0.19 ***) | 0.20 *** (0.22 ***) | 0.20 *** (0.22 ***) | 0.14 *** (0.16 ***) | 0.12 *** (0.14 ***) |
| | HH | 184 (219) | 175 (214) | 175 (215) | 185 (213) | 180 (221) | 129 (165) | 150 (180) |
| | LL | 515 (623) | 536 (656) | 468 (610) | 556 (665) | 601 (698) | 525 (631) | 429 (523) |
| | LH | 95 (105) | 90 (97) | 111 (109) | 96 (95) | 105 (107) | 109 (126) | 111 (121) |
| | HL | 44 (55) | 45 (65) | 31 (51) | 46 (57) | 49 (47) | 48 (60) | 55 (60) |
| | Not sig. | 1639 (1475) | 1631 (1445) | 1692 (1492) | 1594 (1447) | 1542 (1404) | 1666 (1495) | 1732 (1593) |
| 3rd | Moran's I | 0.14 *** (0.18 ***) | 0.11 *** (0.15 ***) | 0.10 *** (0.14 ***) | 0.12 *** (0.17 ***) | 0.13 *** (0.17 ***) | 0.10 *** (0.13 ***) | 0.09 *** (0.11 ***) |
| | HH | 190 (252) | 183 (243) | 173 (251) | 199 (264) | 193 (265) | 150 (229) | 149 (233) |
| | LL | 557 (736) | 591 (740) | 521 (697) | 539 (777) | 628 (824) | 612 (790) | 478 (636) |
| | LH | 127 (138) | 134 (144) | 141 (168) | 123 (143) | 125 (149) | 129 (183) | 126 (171) |
| | HL | 75 (79) | 74 (83) | 62 (69) | 76 (89) | 69 (83) | 74 (87) | 75 (82) |
| | Not sig. | 1528 (1272) | 1495 (1267) | 1580 (1292) | 1486 (1204) | 1462 (1156) | 1512 (1188) | 1649 (1355) |
| 4th | Moran's I | 0.09 *** (0.15 ***) | 0.07 *** (0.12 ***) | 0.06 *** (0.11 ***) | 0.08 *** (0.13 ***) | 0.08 *** (0.14 ***) | 0.07 *** (0.11 ***) | 0.06 *** (0.09 ***) |
| | HH | 181 (260) | 177 (263) | 141 (264) | 171 (313) | 173 (306) | 154 (270) | 146 (261) |
| | LL | 568 (824) | 594 (812) | 526 (756) | 613 (874) | 642 (910) | 613 (845) | 515 (694) |
| | LH | 145 (168) | 153 (185) | 169 (220) | 148 (184) | 154 (202) | 166 (244) | 141 (231) |
| | HL | 72 (93) | 77 (113) | 65 (89) | 75 (107) | 83 (107) | 72 (96) | 82 (112) |
| | Not sig. | 1511 (1132) | 1476 (1104) | 1576 (1148) | 1470 (999) | 1425 (952) | 1472 (1022) | 1593 (1179) |
| 5th | Moran's I | 0.07 *** (0.12 ***) | 0.06 *** (0.10 ***) | 0.05 *** (0.09 ***) | 0.06 *** (0.11 ***) | 0.06 *** (0.11 ***) | 0.05 *** (0.09 ***) | 0.05 *** (0.08 ***) |
| 6th | Moran's I | 0.06 *** (0.11 ***) | 0.05 *** (0.08 ***) | 0.04 *** (0.07 ***) | 0.05 *** (0.09 ***) | 0.06 *** (0.10 ***) | 0.05 *** (0.08 ***) | 0.04 *** (0.07 ***) |
| 7th | Moran's I | 0.05 *** (0.09 ***) | 0.04 *** (0.07 ***) | 0.03 *** (0.06 ***) | 0.05 *** (0.08 ***) | 0.05 *** (0.09 ***) | 0.03 *** (0.07 ***) | 0.04 *** (0.06 ***) |
| 8th | Moran's I | 0.03 *** (0.08 ***) | 0.02 ** (0.06 ***) | 0.02 ** (0.05 ***) | 0.03 *** (0.07 ***) | 0.03 *** (0.07 ***) | 0.02 ** (0.06 ***) | 0.02 ** (0.05 ***) |
| 9th | Moran's I | 0.03 *** (0.07 ***) | 0.02 ** (0.05 ***) | 0.01 ** (0.05 ***) | 0.02 ** (0.06 ***) | 0.03 *** (0.07 ***) | 0.02 ** (0.05 ***) | 0.02 ** (0.05 ***) |
| 10th | Moran's I | 0.03 *** (0.06 ***) | 0.02 ** (0.05 ***) | 0.01 ** (0.04 ***) | 0.02 ** (0.05 ***) | 0.02 ** (0.06 ***) | 0.01 ** (0.04 ***) | 0.02 ** (0.04 ***) |
| 11th | Moran's I | 0.03 *** (0.06 ***) | 0.01 ** (0.04 ***) | 0.01 ** (0.04 ***) | 0.02 ** (0.05 ***) | 0.02 ** (0.05 ***) | 0.01 ** (0.04 ***) | 0.02 ** (0.04 ***) |
| 12th | Moran's I | 0.02 ** (0.05 ***) | 0.01 ** (0.04 ***) | 0.008 * (0.03 **) | 0.01 ** (0.04 ***) | 0.005 * (0.05 ***) | 0.006 * (0.03 **) | 0.01 ** (0.03 **) |
| 13th | Moran's I | 0.01 ** (0.05 ***) | 0.005 ** (0.03 **) | −0.002 (0.03 **) | 0.003 (0.04 ***) | −0.004 (0.04 ***) | −0.008 * (0.03 **) | 0.006 (0.03 **) |
| 14th | Moran's I | 0.006 * (0.04 ***) | −0.001 (0.03 **) | −0.003 (0.02 *) | −0.001 (0.03 **) | −0.007 * (0.03 **) | −0.007 * (0.02 *) | −0.003 (0.03 **) |
| 15th | Moran's I | 0.007 ** (0.04 ***) | 0.007 ** (0.03 **) | −0.001 (0.02 *) | 0.001 (0.03 **) | −0.007 * (0.03 **) | −0.007 * (0.02 *) | −0.004 * (0.02 *) |
| 16th | Moran's I | 0.005 * (0.03 **) | 0.004 (0.02 *) | 0.002 (0.01 *) | −0.003 (0.02 *) | −0.003 (0.03 **) | −0.005 * (0.02 *) | 0.001 (0.02 *) |

**Table A1.** *Cont.*

| Order of Contiguity (Including Lower Order/s) | LISA Clusters/Outliers | DSM2020 | DSM2019 | DSM2018 | DSM2017 | DSM2016 | DSM2015 | DSM2014 |
|---|---|---|---|---|---|---|---|---|
| 17th | Moran's I | −0.001 (0.03 **) | −0.002 (0.02 *) | −0.001 (0.01 *) | −0.01 ** (0.02 *) | −0.01 ** (0.02 *) | −0.01 ** (0.01 *) | −0.002 (0.02 *) |
| | HH | 53 (284) | 38 (279) | 56 (269) | 38 (271) | 37 (251) | 17 (159) | 38 (262) |
| | LL | 514 (1071) | 470 (982) | 424 (922) | 476 (1015) | 518 (1003) | 506 (968) | 393 (927) |
| | LH | 132 (453) | 102 (415) | 120 (465) | 103 (343) | 104 (306) | 91 (321) | 96 (328) |
| | HL | 145 (210) | 152 (211) | 160 (197) | 203 (204) | 238 (207) | 185 (165) | 160 (205) |
| | Not sig. | 1633 (459) | 1715 (590) | 1717 (624) | 1657 (644) | 1580 (710) | 1678 (864) | 1790 (755) |
| 30th | Moran's I | −0.05 *** (0.08 *) | −0.05 *** (0.003 *) | −0.02 ** (0.002 *) | −0.04 *** (0.001 *) | −0.04 *** (−0.001) | −0.02 ** (−0.001) | −0.02 ** (0.003 *) |
| | HH | 67 (216) | 74 (120) | 76 (93) | 74 (59) | 75 (21) | 83 (18) | 91 (146) |
| | LL | 2500 (904) | 221 (975) | 208 (922) | 193 (1034) | 182 (1135) | 163 (1131) | 199 (1052) |
| | LH | 427 (418) | 403 (259) | 369 (177) | 428 (207) | 414 (169) | 362 (181) | 376 (319) |
| | HL | 206 (218) | 179 (266) | 132 (304) | 166 (342) | 139 (405) | 114 (380) | 138 (312) |
| | Not sig. | 1435 (721) | 1508 (857) | 1600 (1041) | 1524 (835) | 1575 (747) | 1663 (767) | 1581 (648) |
| | Neighborless | 92 | 92 | 92 | 92 | 92 | 92 | 92 |

Note: HH—high−high cluster, LL—low−low cluster, HL—high−low outlier, LH—low−high outlier; significance levels: $\alpha$ = * 0.10, ** $\alpha$ = 0.05; *** $\alpha$ = 0.01; DSM—dynamic synthetic measure of organic farming; Not sig.—not significant; the rest of the calculations are available from the author on request.

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
