# Peer review of "Analyzing Spatiotemporal Development of Organic Farming in Poland"

_sustainability, doi:10.3390/su131810399_

Round 1
Reviewer 1 Report
I appreciate this opportunity to review this paper with the title The Detection and Analysis of Spatiotemporal patterns in Eco-agriculture Development: the Case of Poland.
I have some comments
Introduction
It is well done and describes the current state of the research. Some recommendations are only about improving the definition of the research scope.
I found some typos and grammatical errors. English editing is necessary.
Material and methods
They must contain only the materials and methods used in the research in order to allow other researchers to replicate the work.
The paragraphs of the lines 171-181 and 193-204 please move them to the discussion section.
Results and Discussions
We kindly recommend to author that the results be separated from the discussions for a clearer and easier understanding by the readers.
Row 328. Table 1 should be inserted here
Row 364. (p=0.01) specify what it refers to, call the analysis or table
Conclusion
Reflect the results obtained
References
Check all refences and eliminate duplication. For exemple: the reference from line 610 can also be found in line 615
Author Response
Dear Reviewer,
I corrected the paper according to all remarks. Comments were highly insightful and enabled me to greatly improve the quality of the manuscript. Please, find below point-by-point responses to each of the comments.
Comments to the Author
I appreciate this opportunity to review this paper with the title The Detection and Analysis of Spatiotemporal patterns in Eco-agriculture Development: the Case of Poland.
Thank you very much for this statement and motivation.
I have some comments
Introduction
It is well done and describes the current state of the research. Some recommendations are only about improving the definition of the research scope.
I found some typos and grammatical errors. English editing is necessary.
Thank you very much for this remark. I have shortened the Introduction, read it carefully and made some transformations. The paper has been also revised and proofread by the Native Editor (the certificate is attached to the responses).
Material and methods
They must contain only the materials and methods used in the research in order to allow other researchers to replicate the work.
The paragraphs of the lines 171-181 and 193-204 please move them to the discussion section.
Thank you very much for this suggestion. I have transformed the section, i.e. I have separated or moved the paragraphs – some to the Results, and some to the Discussion section of the manuscript.
Results and Discussions
We kindly recommend to author that the results be separated from the discussions for a clearer and easier understanding by the readers.
Row 328. Table 1 should be inserted here
Row 364. (p=0.01) specify what it refers to, call the analysis or table
I agree with all these remarks. I have separated the Discussion from the Results. I have also moved the table into the Results section and explained the p=0.01. The details are included in the Notes: under Figure 2. I have also added additional references and explanations directly in the text.
Conclusion
Reflect the results obtained
Thank you very much for this remark.
References
Check all refences and eliminate duplication. For exemple: the reference from line 610 can also be found in line 615
I agree with this suggestion. I have read the references once more – and removed the duplicates as well as some superfluous notes.

Reviewer 2 Report
Manuscript title: The Detection and Analysis of Spatiotemporal Patterns in Eco-agriculture Development: the Case of Poland
A better title shall be chosen for the manuscript.
Introduction shall be concise – limited to 3 paragraphs including the background, current status and importance along with the objective of the work carried out.
Discussion shall also be limited to 3/ 4 paragraphs and conclusion shall include the main essence of the work along with recommendations or suggestions for the betterment of the agricultural yields.
Author Response
Dear Reviewer,
I appreciate Your decision and I would like to thank You for all the comments. I corrected the paper according to all remarks. I put some accompanying responses on them directly into the paper. Please, find below point-by-point responses to each of the comments.
Comments to the Author
Manuscript title: The Detection and Analysis of Spatiotemporal Patterns in Eco-agriculture Development: the Case of Poland
A better title shall be chosen for the manuscript.
Thank You for this remark. I have changed the title to “Analyzing Spatiotemporal Development of Organic Farming in Poland”. I hope this title is more suitable and adequate for the subject of the study.
Introduction shall be concise – limited to 3 paragraphs including the background, current status and importance along with the objective of the work carried out.
I agree with this suggestion. I have transformed and tighten this paragraph. I have removed some parts and made it more concise.
Discussion shall also be limited to 3/ 4 paragraphs and conclusion shall include the main essence of the work along with recommendations or suggestions for the betterment of the agricultural yields.
I appreciate this remark very much. I have separated the Discussion from the Results and made some transformations to the Conclusions to make them more transparent.
The paper has been also revised and proofread by the Native Editor (the certificate is attached to the responses).

Reviewer 3 Report
I was pleased to know the situation of eco-agriculture in Poland, the authors have done a job that needs to be improved.
Introduction
It is well done and describes the current state of the research. Some recommendations are only about improving the definition of the research scope.
Row 121-127. this detailed explanation is superfluous, it would be enough to conclude the paragraph of the introduction only with the aim of the work.
Materials and methods
They must contain only the materials and methods used in the research in order to allow other researchers to replicate the work. the explanations and comments on the tables, as well as the insertion of the data obtained in this paragraph, are not adequate.
Row 171-181. paragraph M and M should contain only the methods used, this part should be inserted in the discussion paragraph.
Row 193-204. this explanation should be included in the discussions.
Row 210-225. this paragraph needs to be included in the discussions.
Row 226. the table should be moved to the results paragraph
Row 231-259. the comment on the data of table 1 is not part of the paragraph of M and M as it does not describe a method but from what I have read you are discussing the results, so it would be appropriate to transfer everything to the following paragraph.
Results and Discussions
Even if the editor gives the possibility that the paragraph is unique I would prefer that the results be separated from the discussions for a clearer and easier understanding for the reader. The results should provide a precise and concise description of the experimental data obtained, while discussions should encourage the interpretation of the results from the perspective of previous studies.
Row 319-322. it would be advisable for the note to be inserted below line 323 even if it could be inserted without problems in the text.
Row 328. Table 1 should be inserted here.
Row 342. why is table 1 not called table 2? given that it is congruent and well explained of the work done.
Row 364. (p = 0.01) specify what it refers to, call the analysis or table.
Conclusions
Reflect the results obtained
References
References as well as being numbered in order of appearance in the text, should avoid digitization and duplication errors. The DOI must not be inserted because it is not requested by the publisher and it has not always been written in the same form, please delete it. Check all references and eliminate duplication. Example: the row 610 reference would appear to be the same as the row 615.
Author Response
Dear Reviewer,
I appreciate the decision and I would like to thank You for all the previous comments and remarks. I implemented all indicated changes and I hope that the newest revisions in the manuscript will be sufficient to make the study suitable for publication. Please, find below point-by-point responses to each of the comments.
I was pleased to know the situation of eco-agriculture in Poland, the authors have done a job that needs to be improved.
Thank you very much for this recommendation.
Introduction
It is well done and describes the current state of the research. Some recommendations are only about improving the definition of the research scope. Row 121-127. this detailed explanation is superfluous, it would be enough to conclude the paragraph of the introduction only with the aim of the work.
I agree with this suggestion. I have removed the last paragraph from the Introduction. This edition briefed this section.
Materials and methods
They must contain only the materials and methods used in the research in order to allow other researchers to replicate the work. the explanations and comments on the tables, as well as the insertion of the data obtained in this paragraph, are not adequate.
Row 171-181. paragraph M and M should contain only the methods used, this part should be inserted in the discussion paragraph.
Row 193-204. this explanation should be included in the discussions.
Row 210-225. this paragraph needs to be included in the discussions.
Row 226. the table should be moved to the results paragraph
Row 231-259. the comment on the data of table 1 is not part of the paragraph of M and M as it does not describe a method but from what I have read you are discussing the results, so it would be appropriate to transfer everything to the following paragraph.
I would like to thank the Reviewer for these constructive remarks. I have transformed and reconstructed this section according to all the above-mentioned remarks.
Results and Discussions
Even if the editor gives the possibility that the paragraph is unique I would prefer that the results be separated from the discussions for a clearer and easier understanding for the reader. The results should provide a precise and concise description of the experimental data obtained, while discussions should encourage the interpretation of the results from the perspective of previous studies.
Row 319-322. it would be advisable for the note to be inserted below line 323 even if it could be inserted without problems in the text.
Row 328. Table 1 should be inserted here.
Row 342. why is table 1 not called table 2? given that it is congruent and well explained of the work done.
Row 364. (p = 0.01) specify what it refers to, call the analysis or table.
At this point, I fully accept the Reviewer’s opinion. The only I haven’t done was the re-numeration of Table A1 as Table 2 (and inserting it into the body of the text). Table A1 is very extensive, but what is more important – it includes important but a lot of the additional content – being the results of the organic farming spatial autocorrelation with using different spatial weights matrices, and therefore, I have used these outcomes in the selection W (spatial matrix) procedure. So, I have decided to leave Table A1 in the Appendix, however, I have made some notes on it in the text.
Conclusions
Reflect the results obtained
I appreciate this remark very much.
Round 2
Reviewer 2 Report
The authors revised the manuscript sufficiently
Reviewer 3 Report
the authors followed all the suggestions given,
the work was much improved
it's OK for me